# Analysis of Primary Chronic Lymphocytic Leukemia Cells’ Signaling Pathways

**DOI:** 10.3390/biomedicines12030524

**Published:** 2024-02-26

**Authors:** Josipa Skelin, Maja Matulić, Lidija Milković, Darko Heckel, Jelena Skoko, Kristina Ana Škreb, Biljana Jelić Puškarić, Ika Kardum-Skelin, Lipa Čičin-Šain, Delfa Radić-Krišto, Mariastefania Antica

**Affiliations:** 1Ruđer Bošković Institute, 10000 Zagreb, Croatia; josipa.skelin@irb.hr (J.S.); lidija.milkovic@irb.hr (L.M.); darko.heckel@irb.hr (D.H.); jelena.skoko5@gmail.com (J.S.); lipa.cicin-sain@irb.hr (L.Č.-Š.); 2Department of Molecular Biology, Faculty of Science, University of Zagreb, 10000 Zagreb, Croatia; maja.matulic@biol.pmf.hr; 3School of Medicine, University of Mostar, 88000 Mostar, Bosnia and Herzegovina; 4Faculty of Civil Engineering, University of Zagreb, 10000 Zagreb, Croatia; kristina.skreb@gmail.com; 5Department of Clinical Cytology and Cytogenetics, Merkur University Hospital, 10000 Zagreb, Croatia; biljana.jelic.puskaric@zg.t-com.hr; 6School of Medicine, University of Zagreb, 10000 Zagreb, Croatia; ika.kardum-skelin@zg.t-com.hr; 7Department of Internal Medicine, Merkur University Hospital, 10000 Zagreb, Croatia

**Keywords:** AIOLOS, NOTCH, BCL-2, CLL, leukemia, survival

## Abstract

Chronic lymphocytic leukemia (CLL) is a lymphoproliferative disorder characterized by a specific expansion of mature B-cell clones. We hypothesized that the disease has a heterogeneous clinical outcome that depends on the genes and signaling pathways active in the malignant clone of the individual patient. It was found that several signaling pathways are active in CLL, namely, NOTCH1, the Ikaros family genes, BCL2, and NF-κB, all of which contribute to cell survival and the proliferation of the leukemic clone. Therefore, we analyzed primary CLL cells for the gene and protein expression of NOTCH1, DELTEX1, HES1, and AIOLOS in both peripheral blood lymphocytes (PBLs) and the bone marrow (BM) of patients, as well as the expression of BCL2 and miRNAs to see if they correlate with any of these genes. BCL2 and AIOLOS were highly expressed in all CLL samples as previously described, but we show here for the first time that AIOLOS expression was higher in the PBLs than in the BM. On the other hand, NOTCH1 activation was higher in the BM. In addition, miR-15a, miR-181, and miR-146 were decreased and miR-155 had increased expression in most samples. The activation of the NOTCH pathway in vitro increases the susceptibility of primary CLL cells to apoptosis despite high BCL2 expression.

## 1. Introduction

B-cell chronic lymphocytic leukemia (CLL) is the most common chronic blood cancer in adults in Europe. This disease is characterized by an increasing growth of a monoclonal population of cells that are comparable to mature B lymphocytes but functionally incompetent. Similar to the compartment of B-1-like cells, the CLL cells express the CD5 antigen together with CD19, CD20, and CD23 (for a review see [1]). The expression of CD5 on B lymphocytes facilitates the diagnosis of the disease and the characterization of the malignant clone within the samples [2,3]. The origins of the disease are still not clear, but some research suggests that the potential to generate clonal B cells may be acquired in the hematopoietic stem cell stage [4]. CLL can be subdivided into groups depending on whether the cells carry mutations in the variable regions of the immunoglobulin heavy chain genes (IGHV). Patients with unmutated IGHV have been found to present more aggressive disease [5,6]. The two CLL groups can be further subdivided based on the expression of the proteins ZAP70 and CD38, the mutation status of TP53, SF3B1, ATM, and NOTCH1, additional mutations, chromosomal aberrations (deletions of chromosomes 13q, 17p, and 11q as well as trisomy 12), and various important signaling pathways (for a review see [1]). Accordingly, CLL is far from uniform in its clinical presentation as well as the response to therapy, and the aim of our study is to find potential correlations between known pathway markers to improve the prediction of disease progression.

The Ikaros and Notch families have been identified as key regulators of lymphocyte development. Notch signaling is an evolutionarily conserved pathway involved in the differentiation of numerous cell types. The signaling pathway is activated by the binding of a ligand to the Notch receptor, expressed on the plasma membrane of the neighboring cells. This leads to a cascade of proteolytic activity and the final cleavage of the cytoplasmic portion of Notch, (Notch intracellular domain, NICD), which serves as a transcription factor. In hematopoietic cells, some of the NOTCH downstream targets are HES1 and DELTEX1. The expression of ligands and receptors involved in signaling, as well as the signaling strength and outcome, depend on the cells and tissues involved [7,8]. A sequence recognized by the NICD transcriptional cofactor CSL (CBF1, Suppressor of Hairless, Lag-1) could also bind members of the Ikaros family of transcription factors. These proteins are involved in the differentiation of B and T cells. Besides Ikaros, Aiolos is the most prominent member of the family that is expressed in B cells [9,10]. Notch has been connected to T-cell acute lymphoblastic leukemia, where initially found, but its role has also been demonstrated in CLL [11,12]. NOTCH1 was found to be mutated in 6 to 12% of cases at initial diagnosis, while the overall mutation status of patients is more difficult to determine and depends on the time since diagnosis and the stage of disease analyzed [13]. NOTCH1 mutations have been associated with a poor prognosis of CLL and a more difficult to treat form of the disease [14]. Regardless of Notch mutation status, CLL cells constitutively express NOTCH1 and 2, and the active signaling pathway has been found to be involved in apoptosis resistance [12].

Another group of signaling pathways that are deregulated in CLL are those involved in apoptosis. BCL-2 is frequently overexpressed in a number of B-cell malignancies, including CLL, due to the increased transcription or loss of miRNAs that act as its negative regulators [15,16]. CLL cells are generally characterized by in vivo resistance to apoptosis, which remains a major clinical challenge for the successful treatment of the disease [12]. Despite this feature, CLL cells die rapidly when cultured in vitro alone without stromal cells. Co-cultures of leukemic cells with supporting stroma in vitro emphasize the importance of cellular interactions and microenvironmental signals for the proliferation and resistance of neoplastic cells to cell death [17,18].

A number of miRNAs are involved in B cell development and sequentially regulate key genes. They have also been shown to be dysregulated in CLL. The most common are deletions of miR-15A and miR-16-1, which regulate the expression of proteins involved in apoptosis [19]. Deletions of 13q14, the locus on which they are located, are the most common genetic lesion in CLL and occur in up to 60% of patients [20]

The above proteins and microRNAs have their own specific roles in the development, maintenance, and survival of CLL, but none of them is an actor alone; they influence and modify each other. For this reason, we sought to find a link between these prognostic elements and their effects on the survival of malignant clones in a group of CLL patients. The aim of this study was to correlate two or more known factors in an effort to link the existing diagnostic elements and eventually establish their association to cell survival individually. The molecular characteristics of the patients’ cell samples confirmed previous work, but we also found some interesting correlations in the observed factors as well as differences in cell signaling depending on the microenvironment of the cells.

## 2. Materials and Methods

### 2.1. Donor Cells

The reporting of this study conforms to the STROBE guidelines [21]. The study included bone marrow and peripheral blood samples from 20 untreated, newly diagnosed CLL patients and peripheral blood from 3 healthy volunteers. Patients had given written informed consent in accordance with the Helsinki Declaration of 1975 as revised in 2008. The patient samples were anonymized at the hospital and their identity is known only to their physicians. The ethics committees of both the Rudjer Boskovic Institute and the Merkur University Hospital approved this research (approval No. 031-1159314). The average age of the patients was 64 years, and 57% were males. Table 1 shows the clinical characteristics of the patients. They were diagnosed as CLL according to the WHO classification [22]. Clinical stages were assessed according to the Binet and Rai classifications and by means of the quantitative estimation of the size of the tumor mass in 3 main cell compartments (total tumor mass, TTM score) [23], specifically, the number of lymphocytes in the peripheral blood (up to 9 = low risk), the diameter of the largest palpable nodule (9–15 cm = intermediate risk), and the palpable spleen (>15 cm = high risk) (Table 1 and Appendix A). The lymphocytes were separated from whole blood or bone marrow by means of centrifugation on the Lymphoprep™ density gradient medium (Stemcell Technologies, Vancouver, BC, Canada). Due to the amount of material available, the same patient’s sample was analyzed by flow cytometry, protein, and RNA expression.

### 2.2. Quantitative RT-PCR

RNA was extracted with TRIzol™ (Invitrogen, Carlsbad, CA, USA), purity was measured with the BioSPEC-nano Micro-volume UV-Vis spectrophotometer (Shimadzu, Kyoto, Japan), quantified with the Quant-IT RNA Assay Kit (Invitrogen, Carlsbad, CA, USA), and checked by gel electrophoresis. Patient samples with less than 95% CD5^+^CD19^+^ cells and control samples were sorted using the Sony SH800 cell sorter prior to RNA isolation in order to obtain a pure population (Sony Biotechnology, San Jose, CA, USA). Patient samples were sorted by selecting the malignant clone expressing both CD5 and CD19, while control samples were sorted based on CD19 expression, as this is the marker for B cells. RNA was transcribed to cDNA using random hexamers (Roche, Basel, Switzerland) and amplified using SYBR^®^ Select chemistry (Applied Biosystems, Foster City, CA, USA) in an ABI7300 Real-Time PCR System (Applied Biosystems, Foster City, CA, USA). The gene-specific primers were designed manually (Primer-BLAST) [24] and the primers were cited in refs. [25,26]. Ct values were normalized to the housekeeping gene GUSB and this number was used as a negative exponent to base 2 to calculate relative expression. Blood from healthy donors was sorted based on CD19 expression and CD19^+^ cells were analyzed in the same way as the malignant CLL cells.

For miRNA detection, TaqMan Advanced miRNA Assays A25576 and TaqMan MicroRNA Assays PN4427975 (Applied Biosystems, Foster City, CA, USA) with predesigned primers and probes were used. The miRNA expression was normalized to the expression of ribosomal RNA, RNU48, or constitutively expressed miR-432-5p, and presented as relative change compared to a healthy donor.

### 2.3. Flow Cytometry

Multiparameter flow cytometry analysis was performed on patient and control blood samples. Cells were washed in PBS (supplemented with 2% FCS), labeled with primary antibodies for 20 min, washed, and, when appropriate, labeled with a secondary anti-mouse IgG2b Alexa Fluor647 antibody (Invitrogen, Carlsbad, CA, USA) for 20 min. Antibodies to CD5 (TONBO), CD19 (TONBO and eBioscience), BCL-2 (BD), HES1 (Abnova), and NOTCH1 (LSBio) were used [27,28]. Each experiment also included isotype controls. For the detection of intracellular antigens, the cells were first permeabilized with BD Cytofix/Cytoperm™ (BD Biosciences, San Jose, CA, USA) prior to antibody labeling according to the manufacturer’s instructions. Cell viability was determined by propidium iodide (PI) or, when permeabilized, with the LIVE/DEAD^®^ Fixable Red Cell Stain Kit (Molecular Probes, Eugene, OR, USA). PI was added directly to the cell suspension in PBS (supplemented with 2% FCS) immediately before analysis, while the LIVE/DEAD^®^ Fixable Red Dead Cell Stain Kit was used according to the manufacturer’s instructions. Dead cells and debris were excluded from the analysis, and at least 100,000 live cells were processed for each sample. Samples were analyzed on the FACSCalibur (BD Biosciences, San Jose, CA, USA) with 488 and 635 nm lasers. Mean fluorescence intensities (MFI) were calculated with FCS express 3 (De novo Software, Pasadena, CA, USA). The MFI of a corresponding control was subtracted from the sample values to obtain the values shown.

### 2.4. Western Blot Analysis

The lymphocytes of patients with a population of ≥95% CD5^+^CD19^+^ cells, and reflecting the composition of the study population by sex and Binet and Rai classification, were selected and processed as previously described [9,28]. Briefly, 5 µg of proteins from each sample were separated using SDS-PAGE on a 10% gel, transferred to a nitrocellulose membrane, blocked in Blotto buffer for one hour, and incubated overnight with primary antibodies: Rabbit anti-human AIOLOS, Rabbit anti-human cleaved NOTCH1, Rabbit anti-human NOTCH1, and Rabbit anti-human β-ACTIN (all from Cell Signaling Technology, Inc., Beverly, MA, USA) [27,28]. After washing, the horseradish peroxidase (HRP)-linked anti-rabbit antibody was applied for one hour and the signal was detected by chemiluminescence using Supersignal West Pico plus (Thermo Fisher Scientific, Waltham, MA, USA) on a UVITEC Imaging System (Cleaver Scientific, Rugby, UK).

### 2.5. Cell Lines and Co-Culture Experiments

OP9-GFP cells or OP9-DL1 cells were a kind gift from Professor J.C. Zuniga-Pflücker (Department of Immunology, University of Toronto) [29]. The cell lines were cultured in α-MEM medium (Thermo Fisher Scientific, MA, USA) supplemented with 20% FCS and L-glutamine [29]. CLL cells (2 × 10^5^) from four different patients (No. 2, 3, 10, and 13) were cultured on non-confluent OP9-DL1 or on control OP9-GFP cells, and incubated for 2, 5, and 7 days. In parallel, CLL cells were cultured in medium alone. Cells were harvested and incubated with CD45 APC (Tonbo Biosciences, San Diego, CA, USA) to determine the lymphocyte population, and with Annexin-V-FLUOS (Roche, Basel, Switzerland) and propidium iodide for apoptosis analysis by flow cytometry (FACSCalibur™ Flow Cytometer, Beckton Dickinson, Franklin Lakes, NJ, USA, and Sony Biotechnology SH800, San Jose, CA, USA).

### 2.6. Statistics

Statistical analyses were performed using GraphPad Prism version 7.05 for Windows (GraphPad Software, La Jolla, CA, USA). Mean values between two groups were compared using a two-tailed, unpaired Student’s *t*-test or Mann–Whitney U-test, as appropriate. The relationship between parameters was evaluated by the Pearson correlation coefficient. The results are presented as individual values and/or group means with standard deviation (SD) or standard error of the mean (SEM) as indicators of variability. Differences were considered statistically significant if *p* < 0.05.

## 3. Results

We performed a comprehensive multiparameter analysis of 20 newly diagnosed CLL patients at the gene and protein expression level. Pathways that are frequently deregulated in CLL, such as Notch and molecules that interfere with it, as well as the anti-apoptotic Bcl-2 were analyzed. In addition, we also analyzed the expression of several important miRNAs for leukemia development.

### 3.1. NOTCH1 Activation and AIOLOS Expression in CLL Patients

As the Notch signaling pathway is dysregulated in a certain number of CLL patients, we analyzed the activity of this pathway in the peripheral blood lymphocytes (PBL) of 18 CLL patients. Flow cytometry was used to analyze the expression of three proteins: cleaved NOTCH1 produced by pathway activation; HES1, a downstream target of the Notch signaling; and AIOLOS, a member of the Ikaros family known to interfere with the Notch pathway and found to be deregulated in B-CLL. Cleaved Notch was detected in 67% (12/18) of samples, HES1 was elevated in 39% (7/18), and AIOLOS was elevated in 33% (6/18) of samples compared to CD19^+^ normal B lymphocytes levels (Figure 1).

We also analyzed the gene expression profile of the CLL samples using qPCR (Figure 2). The expression of NOTCH1, its downstream targets DELTEX1 and HES1, and the expression of AIOLOS in CLL blood samples were compared with the expression of CD19^+^ normal B lymphocytes from peripheral blood. NOTCH1 expression was elevated in 38% of samples, DELTEX1 in 33%, and HES1 in 19%. AIOLOS was increased in 71% of the samples.

We correlated the expression of genes and proteins in the individual patients. The information used were the 2^−ΔCT^ values (Figure 2) for the gene expression correlation and the median fluorescence intensity obtained in flow cytometric analyses (Figure 1) for protein expression correlation. The Western blot analysis is in Appendix A. Samples from four patients (No. 3, 5, 15, and 18) who were characterized by lymphocytosis, enlarged lymph nodes, spleen, or liver, and low erythrocyte and platelet counts, and belonged to the high-risk group (stage C or IV) according to both the Binet and Rai staging systems were excluded from the analysis. Two of these high-risk samples (5 and 15) when included significantly affected the correlation statistics. The analysis of the correlations between NOTCH1 and DELTEX1 and NOTCH1 and HES1 revealed correlation coefficients of r = 0.678 and r = 0.709, respectively. The increase in the relative gene expression of NOTCH1 was accompanied by an increased expression of DELTEX1 or HES1 in CLL patients, confirming the activity of the pathway. On the other hand, the relative gene expression of NOTCH1 correlated only moderately with AIOLOS expression (r = 0.557), but this correlation was also confirmed at the protein level (*p* < 0.0001) (Figure 3).

### 3.2. BCL2 Expression in CLL Patients

To better examine the resistance of CLL to apoptosis, we analyzed the expression of the anti-apoptotic protein BCL-2 in 10 CLL samples using flow cytometry. BCL-2 expression levels were increased in all 10 malignant CLL clones (CD5^+^CD19^+^) compared to other lymphocyte populations (CD5^+^ or CD19 single-positive single cells) of the CLL samples. We also compared the expression level of BCL-2 in CLL patients with the level of lymphocyte populations in the peripheral blood cells of a healthy individual. In the blood samples of CLL patients, we detected increased BCL2 levels in both CD5^+^CD19^+^ cells, typical for CLL, and in the CD19^+^ positive single cells (Figure 4).

### 3.3. Oncogenic and Tumor Suppressor miRs Are Deregulated in CLL

Since miRNA expression is frequently deregulated in B-CLL cells, seven CLL and one healthy donor sample were analyzed for the relative expression of miR-7, miR-34a, miR-15a, miR-155, miR-181, miR-29a, and miR-146. The expression of miRs was correlated with the expression of ribosomal RNA or constitutively expressed miR-432-5p (for miR-181, miR-155, miR-15a, miR-34a, and miR-7), and the relative values of miR expression are shown in Figure 5. In the majority of CLL samples, we detected a loss of miR-15a and low expression of miR-181 and miR-146. Similarly, miR-155, the CLL progression marker, was upregulated in six out of seven CLL samples. The tumor suppressor miR-29a was downregulated in aggressive CLL in contrast to indolent CLL. The expression of miR-7 was not significantly altered in the majority of samples. Finally, miR-34a, a tumor suppressor that acts downstream of activated p53 and downregulates ZAP70, was significantly upregulated in four of the CLL samples (nos. 1, 5, 10, and 13), which belong to the intermediate risk group according to TTM scoring [30].

### 3.4. Notch Pathway Activity in Bone Marrow and Peripheral Blood Samples

To investigate the influence of the cell microenvironment on Notch signaling, ten B-CLL samples were analyzed using qRT-PCR for the expression of the NOTCH1, DELTEX1, HES1, and AIOLOS genes in both peripheral blood and bone marrow. AIOLOS expression was significantly higher in peripheral blood, while NOTCH1 expression was similarly high in both compartments. Although DELTEX1 and HES1 were more highly expressed in the bone marrow, their expression was very low in both groups. In addition, the expression levels of Hes1, cleaved Notch, and Aiolos, obtained using flow cytometry, were compared. At the protein level, the expression of Aiolos was similar in peripheral blood and bone marrow, while the expression of cleaved Notch was significantly higher in the bone marrow. Although Hes1 also appeared to be more highly expressed in bone marrow, the data did not differ significantly due to the high inherent variability of the samples (Figure 6).

### 3.5. In Vitro CLL Cell Survival

Considering that CLL rapidly undergoes apoptosis in liquid culture, we decided to use the OP9 cell line transfected with the NOTCH ligand DLL-1. The OP9 cells were used to test whether the activation of the Notch signaling pathway would rescue the cells from apoptosis. We seeded primary CLL lymphocytes originating from CLL patients on OP9 cells expressing the ligand DLL1 (OP9-DL1) or on control OP9 cells expressing GFP (OP9-GFP). As a second negative control, we cultured the CLL cells in the medium alone. The co-cultures were maintained for 2, 5, and 7 days (Figure 7). To exclude the detached OP9 cells from the analysis, the samples were gated for CD45-positive cells. The representative dot plots showing the apoptosis rate in CD45^+^ cells are shown in Figure 7A,B. When cultured with either OP9-DL1 or OP9-GFP cells, the overall survival rate of CLL cells was significantly increased compared to cultures in medium alone where on day 7 most cells were dead (Figure 7C). The comparison of CLL cultures grown on OP9-GFP and OP9-DL1 cells showed a similar percentage of apoptotic cells (Figure 7D). However, the comparison of the absolute number of cells showed that more cells survived on OP9-GFP than on OP9-DL1 cells (Figure 7E), indicating that the Notch signaling pathway negatively affects cell survival despite the high BCL-2 expression in CLL cells.

## 4. Discussion

The process of differentiation is seen as a coordinated cascade of changes in the cell signaling network. This process is particularly complex in hematopoietic cells, and the development of leukemia is thought to be due to defects in the regulation of the differentiation signaling process. The signaling pathways involved in the regulation of proliferation during this process cooperate with the signaling pathways involved in apoptosis, differentiation, and other processes via numerous feedback loops. In leukemia cells, multiple processes are likely to be impaired in parallel, as the cell attempts to maintain normal function and compensate for the signaling failures at the onset of the initial changes through various mechanisms. Although all signaling pathways in the cell are somehow interconnected, CLL cells from each patient exhibit a different combination of dysregulated pathways [1,30]. To determine whether there are parallels between the overactivated signaling pathways in CLL samples, we analyzed and correlated several commonly activated signaling circuits. We analyzed the expression of the transcription factor AIOLOS, which is frequently elevated in CLL, the activity of the Notch pathway, which is involved in cell proliferation, and a pathway involved in resistance to apoptosis and chemotherapy, namely, the expression of BCL-2. In addition, we analyzed the expression of several miRNAs, which also modulate important signaling pathways and may act as tumor suppressors or oncogenes in CLL.

In our previous work on CLL, we focused our attention on modulators of lymphocyte differentiation such as members of the Ikaros family of zinc finger proteins [18]. These transcription factors modify the expression of genes by binding and remodeling chromatin, thereby regulating nuclear activity and controlling the lineage specification of B cells [31]. AIOLOS, a member of the Ikaros family, is essential for the normal development of B cells. AIOLOS has been found to be highly expressed in CLL cells, possibly due to NF-κB signaling activity, which causes a change in the epigenetic regulation of the AIOLOS promoter. On the other hand, it was found that Aiolos mutations led to CLL-like disease in mice and transcriptionally activate genes involved in BCR-NF-κB signaling. Thus, it can lead to an increase in BCR signaling, which is considered to be the main pathway deregulated in CLL. The overexpression of AIOLOS has been associated with the expression of anti-apoptotic members of the Bcl-2 family, particularly BCL-2, and leukemic cell survival [9,10,32]. On the other hand, this increased expression does not correlate with IgVH status, ZAP-70 expression, or CD38 expression, and AIOLOS isoforms are normally distributed between the nucleus and cytoplasm of CLL cells [33]. In this study, the CLL samples showed increased AIOLOS mRNA levels. Furthermore, we found a discrepancy between the mRNA and protein levels of AIOLOS, which could indicate the activity of upstream signaling pathways that drive AIOLOS expression, but also post-transcriptional control mechanisms that prevent its translation. AIOLOS expression correlated with Notch pathway activity, and both AIOLOS protein expression and NOTCH1 activation were higher in BM than in PBL. The Ikaros family and Notch signaling pathways converge at the level of transcriptional regulation and have been shown to work together under certain conditions [33]. We can speculate that the bone marrow microenvironment controls the leukemic clone via the NOTCH1 loop, but when AIOLOS levels and thus NOTCH1 decrease, the cells leave the bone marrow and migrate to the peripheral blood where they become an indolent, long-lived malignant clone. The increased survival of CLL cells in the bone marrow microenvironment has also been confirmed by the co-culture of CLL cells on stromal cells [34,35]. Experiments also showed that blocking individual Notch receptors and ligands in culture experiments leads to a reduction in CLL cell survival in all cases except for DLL-1 and NOTCH3, where no statistically significant difference was found [36,37]. Previous work by other authors has shown that active Notch signaling improves cell survival in vitro [11]. However, the co-culture experiment described here shows that cell survival was worse when the malignant CD5^+^CD19^+^ clone was re-cultured with the cell line expressing the NOTCH ligand DLL1 and mimicking the in vivo bone marrow environment (compared to co-culture with the control OP-9 cells). These data suggest that the activity of the Notch signaling pathway may negatively impact CLL cell survival in the bone marrow. This suggests that the malignant clone cannot be rescued by induced Notch signaling alone and cannot be restored to the bone marrow state. On the other hand, recent experiments have shown that the survival of CLL cells increases in co-culture with macrophages as nurse cells, and it appears that the microenvironment in the bone marrow, as well as in the blood, produces various growth factors that enable cell survival [34].

Furthermore, the confirmed high BCL2 expression level had no effect on the onset of apoptosis in vitro. Interestingly, when we examined BCL2 expression in the malignant clone, we found that the residual normal B cells from CLL patients also expressed higher levels of BCL2. Since CD19 is expressed on all cells of the B lineage and B lymphocytes express different levels of BCL2 during their development [38], we could not pinpoint whether this increase was due to a specific subpopulation overexpressing BCL2 or a general phenomenon, but it is certainly an interesting topic for further investigation.

To better characterize individual patients, we examined a number of microRNAs, some of which were already known to be prognostic markers for CLL. We also wanted to compare the expression of these microRNAs with that of our analyzed genes and proteins, but found no specific correlation between them. The miR-15a/16b cluster was found to be frequently deleted or downregulated in CLL [39]. Its key function is to downregulate anti-apoptotic Bcl2 and Mcl1 and inhibit proliferation [40]. miR-15a was downregulated in our cohort, which may be due to the small number of patients studied. miR-155 is considered an oncomir and marker of CLL progression, with increasing expression paralleling the progression from normal B cells to monoclonal B lymphocytosis and chronic lymphocytic leukemia [41]. miR-155 was upregulated in CLL, which is interesting, as studies show that NOTCH expression negatively regulates miR-155, while miR-155 increases NF-κB activation by targeting its inhibitor [42,43]. Our results confirm Ferrajoli’s findings, as all CLL samples examined showed increased levels of miR-155. miR-29a and miR-181 negatively regulate the expression of TCL1, an oncogene whose high expression is found in aggressive forms of CLL [44], and also downregulated BCL-2 [45,46]. Although they have a similar mechanism of action, they behaved differently in our patient group. miR-181 was downregulated, while the expression of miR-29a depended on whether the patient in question had aggressive or indolent CLL. This finding is consistent with sparse previous results showing that miR-29a is expressed in indolent CLL [44,47]. Since miR-146 is an important immune regulator, especially in the myeloid lineage, we wanted to find out whether it is dysregulated in CLL. miR-146 was downregulated in all our samples. In an analysis of a larger number of CLL samples, no significant difference between the expression of miR-146 in CLL and normal B-cell subpopulations was previously observed. Interestingly, however, expression in CLL was dependent on IGHV mutation status [48]. miR-7 and miR-34a are both considered tumor suppressors and are induced by p53 following DNA damage. Among other pathways, both are thought to positively influence apoptosis by targeting BCL-2 [49]. miR-34a has previously been found to be downregulated in CLL patients with a 17p (TP53) mutation and can be induced by irradiation [50], but here we found a striking upregulation in the majority of our untreated patients (4/7). In contrast, miR-7 is only slightly upregulated in a minority of patients and cannot be correlated with miR-34a expression. BCL2 expression in the malignant clone remained fairly constant regardless of miR-7 or miR-34a expression, while AIOLOS and NOTCH1 expression did not follow any particular pattern.

As Croatia is a small country with a typical annual CLL incidence for Europe [51], the collection and analysis of a large cohort is, unfortunately, beyond the scope of this paper. The number of patients examined in this study limits the ability to draw firm conclusions about the interplay of the different proteins and miRNAs described here. Nevertheless, our work provides useful insights and could be a valuable reference for groups that have access to a larger number of patients [52,53]. The discoveries of the correlation of AIOLOS and NOTCH on the protein level, as well as the differing levels of these two proteins in the bone marrow and peripheral blood, are particularly interesting and should be explored in a larger cohort in order to determine their biological significance. Considering the importance of both factors in B-cell differentiation and CLL development, continuing this line of research should provide important insights into the dynamics of the progression of this malignancy. Specifically, the co-cultures we established with OP9-DL1 provide an in vitro functional tool to test the CLL microenvironment interaction and possible inhibitors’ treatment strategies.

## 5. Conclusions

In summary, we found that the expression of AIOLOS and NOTCH1 in the cells of the malignant clone may vary depending on their location in the body and that their expression may be inversely correlated. We confirm, as previously shown, that CLL cells exhibit an increased expression of the anti-apoptotic BCL2 as well as a dysregulation of miRNAs involved in the development or maintenance of CLL. However, we show here that despite the increased BCL2 expression in CLL cells, Notch activation induced by the microenvironment in vitro causes them to undergo apoptosis. The aim of our work was to establish the cell co-cultures for Notch activation to test the susceptibility of cells to apoptosis in individual patients, and to target the CLL microenvironment interaction which could be a valuable tool for further investigation and consequently for diagnostic or treatment indications.

## Figures and Tables

**Figure 1 biomedicines-12-00524-f001:**
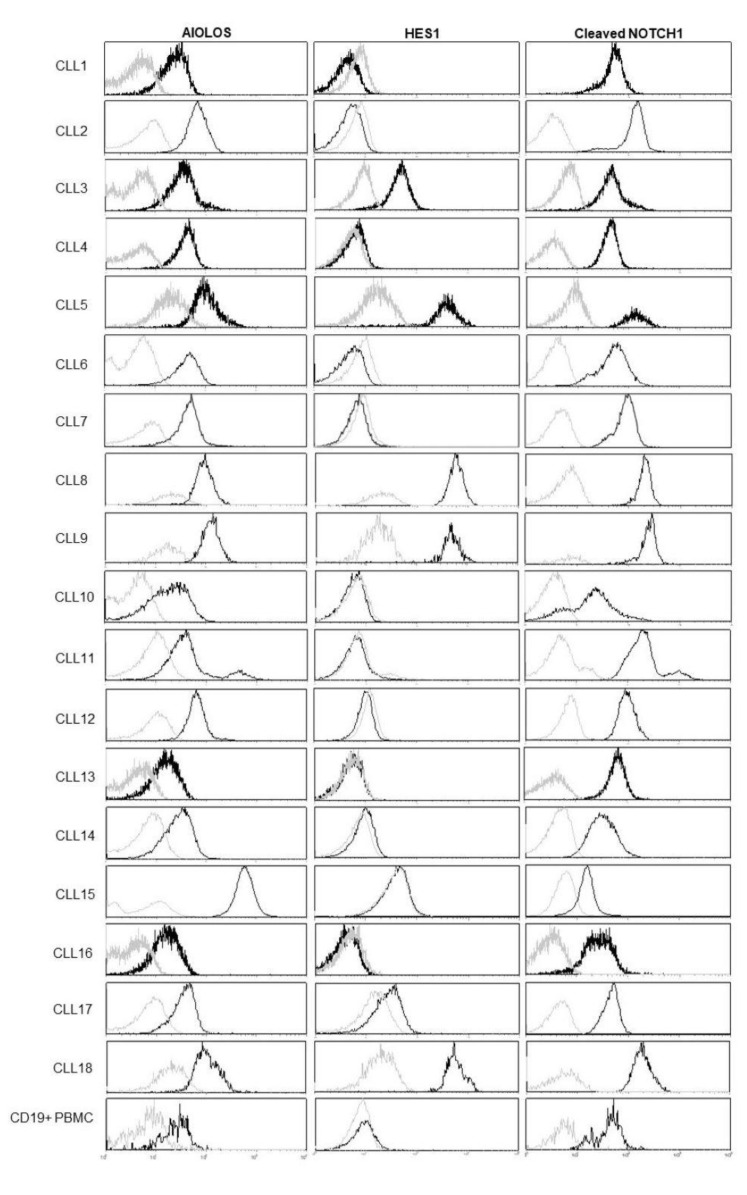
Multiparameter flow cytometry analysis of AIOLOS, HES1, and cleaved NOCTH1 in the malignant CLL clone and in the CD19^+^ peripheral blood lymphocytes. Dead cells were excluded from the analysis. A total of 18 CLL samples and control cells were incubated with the corresponding antibodies and analyzed using multiparametric flow cytometry. Light lines: control samples; dark lines: CLL clones.

**Figure 2 biomedicines-12-00524-f002:**
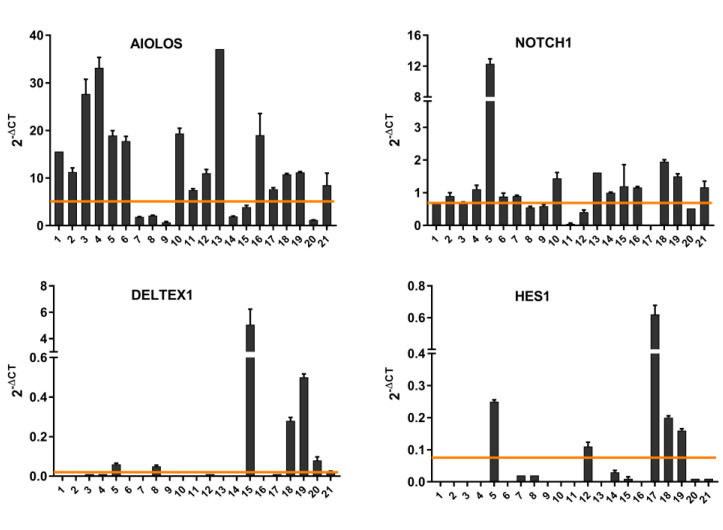
qPCR analysis of CLL samples for the mRNA expression of NOTCH1, DELTEX1, HES1, and AIOLOS. cDNA was prepared from RNA isolated from blood samples of CLL patients and compared with normal CD19^+^ peripheral B cells. Relative expression levels were normalized with GUS gene expression. Data are expressed as 2^−ΔCT^. Mean values and SEM of replicates are shown. The values of the control samples are shown as a solid line.

**Figure 3 biomedicines-12-00524-f003:**
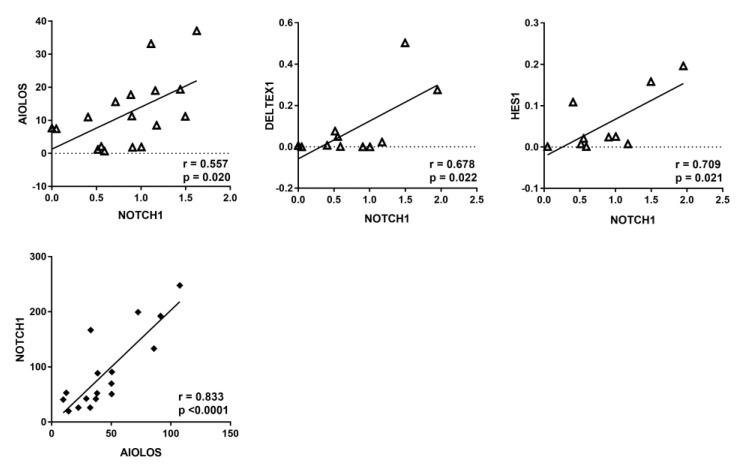
Correlation plot of data from gene expression (**top**) and protein analysis (**bottom**) of CLL samples for NOTCH1, HES1, DELTEX, and AIOLOS. Correlation coefficients and *p*-values are indicated.

**Figure 4 biomedicines-12-00524-f004:**
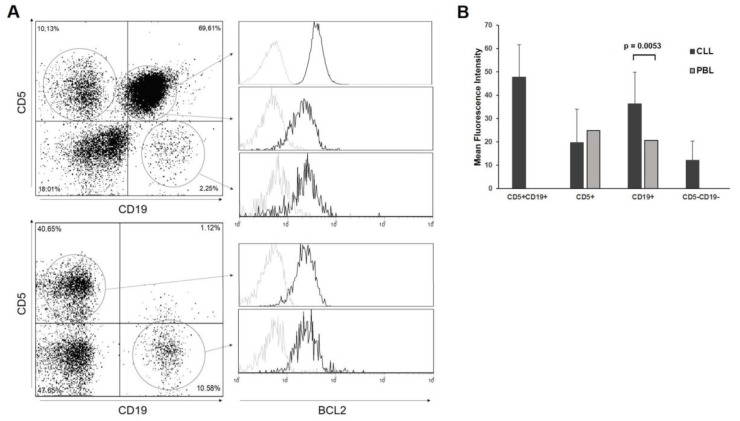
(**A**) Analysis of BCL2 expression in a representative CLL blood sample (top) and in a peripheral blood sample from healthy control (bottom) using flow cytometry. The cell subpopulations according to the expression of CD5 and CD19 were analyzed for BCL2 expression. (**B**) Mean fluorescence intensity for each population calculated by normalizing the MFI of a gene with a corresponding isotype control. The bars represent the mean and SEM of 10 patients’ samples. The *p*-value is indicated in the figure. BCL2 expression (black line) and the according isotype control (gray line).

**Figure 5 biomedicines-12-00524-f005:**
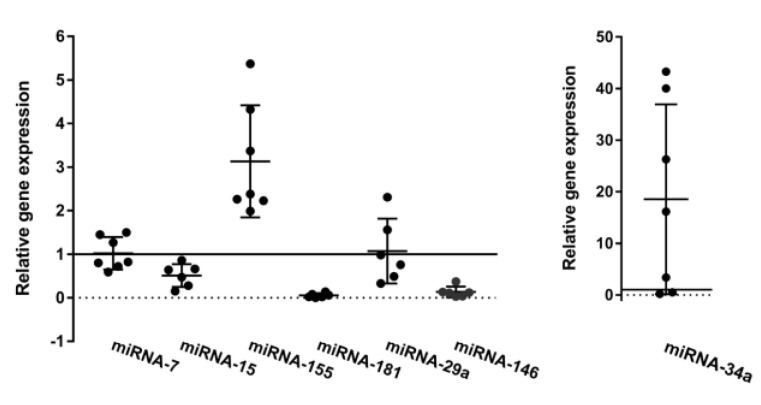
miRNA expression detected with TaqMan Advanced miRNA Assays. Relative gene expression for miR-7, miR-34a, miR-15, miR-155, miR-181, miR29a, and miR-146 for seven independent B-CLL samples (patients no 1, 3, 4, 5, 10, 13, and 16) is shown as individual values with mean ± SD indicated. The values of the control samples are shown as a solid line at value 1.

**Figure 6 biomedicines-12-00524-f006:**
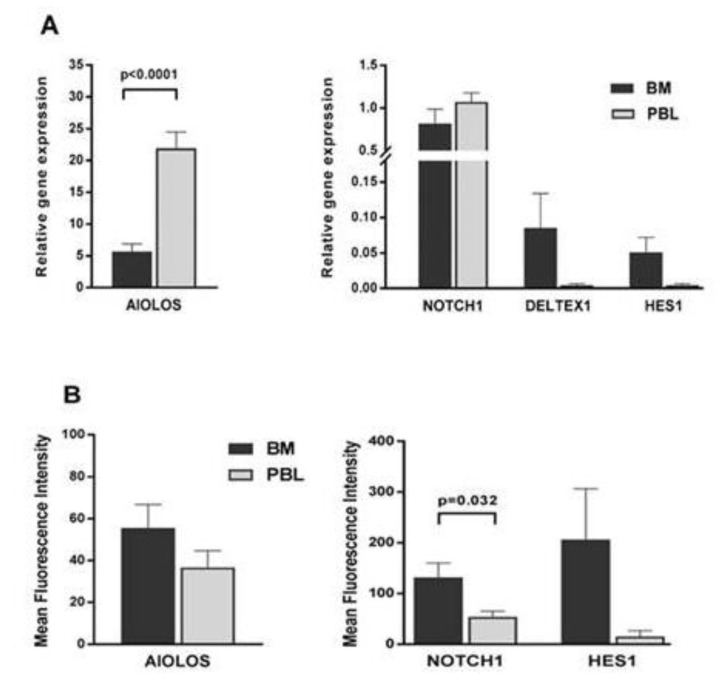
Gene and protein expression profile of primary leukemia samples in peripheral blood and bone marrow. (**A**) Ten B-CLL cells, collected from peripheral blood (PBL) and bone marrow (BM), were analyzed for NOTCH1, DELTEX1, HES1, and AIOLOS gene expression using qRT-PCR. Ct values were normalized to the expression of the GUSB housekeeping gene, and expressed as relative gene expression. (**B**) The protein expression of NOTCH1, HES1, and AIOLOS was analyzed using flow cytometry after the labelling of cells with the corresponding antibodies.

**Figure 7 biomedicines-12-00524-f007:**
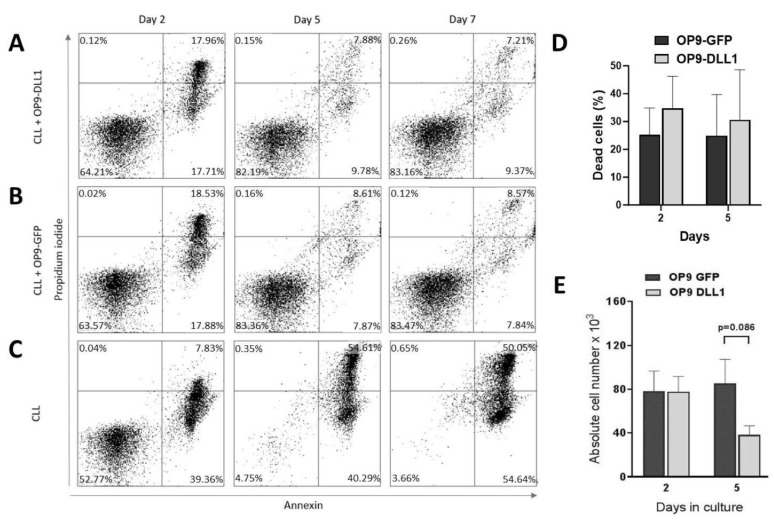
(**A**–**C**) Representative results of the flow cytometric analysis of cell death of a CLL sample cultured on an OP9-DL cell layer (**A**), an OP9-GFP cell layer (**B**), or in medium alone (**C**). To exclude the detached OP9 cells from the analysis, the samples were gated for CD45-positive cells. (**D**) Comparison of the percentage of dead (apoptotic and necrotic) cells recovered from the cultures described in A and B. The bars represent the mean ± SEM of four experiments. (**E**) Comparison of the absolute number of cells recovered from cultures described in (**A**,**B**). The bars represent the mean ± SEM of four experiments.

**Table 1 biomedicines-12-00524-t001:** Clinical and laboratory data of CLL patients diagnosed according to the WHO classification, including Binet, Rai, and TTM staging. Mean and range of values (in parenthesis) are given throughout. TTM, total tumor mass.

CLL Classification	Stage	No. of Patients	Age, Years	Leukocytes (×10^9^/L)	Lymphocytes (×10^9^/L)	Neutrophils (×10^9^/L)	Thrombocytes (×10^9^/L)	Hemoglobin (g/L)
Binet	A	8	64 (37–77)	55 (11–107)	47 (5–99)	4.6 (2–7)	203 (111–304)	139 (127–160)
	B	6	61 (45–70)	68 (19–250)	62 (12–262)	3.2 (2–5)	214 (128–491)	120 (106–140)
	C	6	66 (54–75)	189 (23–645)	175 (20–612)	7.0 (1–13)	125 (43–262)	88 (73–117)
Rai	0	2	69 (63–74)	41 (11–70)	30 (5–54)	5.0 (5–5)	251 (198–304)	130 (127–133)
	I	5	57 (37–75)	45 (15–107)	40 (11–99)	4.0 (2–5)	204 (183–228)	142 (130–160)
	II	5	69 (65–77)	48 (15–85)	42 (12–78)	4.0 (2–7)	147 (119–166)	129 (112–146)
	III	4	64 (45–75)	157 (46–250)	149 (42–242)	5.3 (2–8)	269 (152–491)	91 (67–111)
	IV	4	63 (54–69)	200 (23–645)	184 (20–612)	6.8 (1–13)	80 (43–98)	95 (73–117)
TTM	0–9	8	61 (37–75)	41 (11–107)	35 (5–99)	3.9 (1–5)	203 (98–304)	133 (92–160)
	10–15	7	69 (63–77)	64 (15–92)	55 (12–83)	5.3 (2–8)	145 (85–262)	118 (81–146)
	>15	5	60 (54–70)	240 (20–645)	227 (13–612)	6.0 (2–13)	203 (43–491)	94 (67–112)

## Data Availability

All additional data associated with the paper is available upon request to interested researchers.

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
