# Peer review of "Analysis of Primary Chronic Lymphocytic Leukemia Cells’ Signaling Pathways"

_biomedicines, 2024, doi:10.3390/biomedicines12030524_

Round 1

Reviewer 1 Report

Comments and Suggestions for Authors

The authors analyzed the expression of several genes at the RNA and protein levels in cells from patients with chronic lymphocytic leukemia.  The originality and novelty of the data is very limited, this work is rather of confirmatory nature.

Specific Points of Criticism and Suggestions for Alterations:

(1)  Authorline:  Which of the authors is associated with Napoli, Italy (superscript #6)?

(2)  English:  In general the English of the manuscript is acceptable and understandable; however, a rigorous editing would certainly improve the text.  One example:  „outliers“ instead of „outliners“ (line 228).

(3)  Table 1:  This large table could go into the supplements.

(4)  Percentages:  It is sufficient to present the percentages with whole numbers without the decimals (for example on line 220 but also elsewhere:  38% instead of 38.1% and 33% instead of 33.3% etc.).

(5)  Figure 2:  A horizontal line could be included in the four graphs which indicates when an expression is deemed to be significantly increased.

(6)  Figure 5:  Is there any order how the miRNAs are arranged? (for example according to ascending numbers:  miRNA-7, -15, -29a, -146, -155, -181).

(7)  Figure 7 and paragraph 3.4:  There are no actual data in this figure (e.g. percentages).

(8)  Lines 421-422:  Croatia is in the European Union. The authors might want to be involved in cooperations with other European investigators and might want to be involved in „CLL networks“.

(9)  Any outlook? What to do next?

Comments on the Quality of English Language

Moderate editing suggested

Reviewer 2 Report

Comments and Suggestions for Authors

In this manuscript entitled “Analysis of primary chronic lymphocytic leukemia cells' signaling pathways”, the authors analyzed the expression of NOTCH1 activation, AIOLOS expression, BCL2 expression, miRNAs expression, Notch pathway activity, and detected cell apoptosis after the coculture of CLL cells and NOTCH ligand DLL-1 transfected cell line OP9.

1. First, the theme of this study is unclear and lack of novelty. The signaling pathways are heterogeneous and complex in CLL, but the authors only checked the Notch signaling pathway and BCL2 expression already showed significant in CLL.

2. Moreover, the references are old and some conclusions summarized in the introduction doesn’t match the corresponding references such as the references #7 and #8 in Line 59, Page 2, and the reference #12 in Line 70, Page 2.

3. Next, the methods in this study are simple and unilateral. In Figure 3, the authors confirmed the correlation of NOTCH1 expression and AIOLOS expression on the protein level, but the figures of Western Blot are not provided and lack of a series of Notch function loss or acquisition experiment for validation.

4. In addition, the study is not well organized and lack of logic. The five parts of results seem to be simply combined together, but there is no clear flow of the story.

5. Finally, there are some spelling mistakes in the manuscript.

Thus, this paper is not acceptable for publication in its present form.

Comments on the Quality of English Language

Moderate editing of English language required.

Reviewer 3 Report

Comments and Suggestions for Authors

In the current manuscript, the authors analyzed activated signaling pathways of chronic lymphocytic leukemia using patient cells. The results are interesting, and the manuscript is almost well written. Here are my comments that would improve the manuscript.

1.       Although the current study shows the heterogeneity of NOTCH1 activity by protein and mRNA expressions, it is unclear whether the cases with NOTCH1 activation possessed NOTCH1 and the related genomic mutation or not. The information will be helpful for the readers to understand the current results. This information also will be important for assessment of co-culture model with OP9 cells in Figures 7. In Figures 7, which cells did the authors use for the co-culture model?

2.       Introduction and Discussion sections are needed to be more summarized.

Minor concerns:

1.       What does “[for review 1]” mean (Page 1, Line 35)?

2.       Is SF3B1 correct (Page 1, Line 44)?

3.       It may be better to summarize clinical and pathological findings in a summarized table and the current detailed data would be informative as a supportive data.

Round 2

Reviewer 3 Report

Comments and Suggestions for Authors

The authors provided several data and modified the manuscript according to the Reviewer's comments. Those changes have made the manuscript clearer.